# High Glucose Reduces the Paracellular Permeability of the Submandibular Gland Epithelium via the MiR-22-3p/Sp1/Claudin Pathway

**DOI:** 10.3390/cells10113230

**Published:** 2021-11-19

**Authors:** Yan Huang, Hui-Min Liu, Qian-Ying Mao, Xin Cong, Yan Zhang, Sang-Woo Lee, Kyungpyo Park, Li-Ling Wu, Ruo-Lan Xiang, Guang-Yan Yu

**Affiliations:** 1National Engineering Laboratory for Digital and Material Technology of Stomatology and Beijing Key Laboratory of Digital Stomatology, National Clinical Research Center for Oral Diseases, Department of Oral and Maxillofacial Surgery, Peking University School and Hospital of Stomatology, Beijing 100081, China; hynh@bjmu.edu.cn (Y.H.); maoqianying0418@126.com (Q.-Y.M.); 2Key Laboratory of Molecular Cardiovascular Sciences, Ministry of Education, Beijing Key Laboratory of Cardiovascular Receptors Research, Department of Physiology and Pathophysiology, Peking University School of Basic Medical Sciences, Beijing 100191, China; 1811210002@pku.edu.cn (H.-M.L.); congxin634@126.com (X.C.); zhangy18@bjmu.edu.cn (Y.Z.); pathophy@bjmu.edu.cn (L.-L.W.); 3Department of Physiology, School of Dentistry, Seoul National University, Seoul 110-749, Korea; goodman2324@naver.com (S.-W.L.); kppark@snu.ac.kr (K.P.)

**Keywords:** diabetes, submandibular gland, microRNA-22-3p, tight junction, specificity protein-1

## Abstract

Tight junctions (TJs) play an important role in water, ion, and solute transport through the paracellular pathway of epithelial cells; however, their role in diabetes-induced salivary gland dysfunction remains unknown. Here, we found that the TJ proteins claudin-1 and claudin-3 were significantly increased in the submandibular glands (SMGs) of db/db mice and high glucose (HG)-treated human SMGs. HG decreased paracellular permeability and increased claudin-1 and claudin-3 expression in SMG-C6 cells. Knockdown of claudin-1 or claudin-3 reversed the HG-induced decrease in paracellular permeability. MiR-22-3p was significantly downregulated in diabetic SMGs and HG-treated SMG-C6 cells. A miR-22-3p mimic suppressed claudin-1 and claudin-3 expression and abolished the HG-induced increases in claudin-1 and claudin-3 levels in SMG-C6 cells, whereas a miR-22-3p inhibitor produced the opposite effects. Specificity protein-1 (Sp1) was enhanced in diabetic SMGs and HG-treated SMG-C6 cells, which promoted claudin-1 and claudin-3 transcription through binding to the corresponding promoters. A luciferase reporter assay confirmed that miR-22-3p repressed Sp1 by directly targeting the Sp1 mRNA 3′-untranslated region (3′-UTR). Consistently, the miR-22-3p mimic suppressed, whereas the miR-22-3p inhibitor enhanced, the effects of HG on Sp1 expression. Taken together, our results demonstrate a new regulatory pathway through which HG decreases the paracellular permeability of SMG cells by inhibiting miR-22-3p/Sp1-mediated claudin-1 and claudin-3 expression.

## 1. Introduction

Saliva is vital to oral and systemic health, as it contributes to mechanical cleaning and plays a protective role in the oral cavity through physiological and biochemical mechanisms [1]. Patients with diabetes exhibit a high prevalence of salivary gland dysfunction, which disturbs the homeostasis of the oral cavity by altering the salivary composition and volume [2,3]. Hyposalivation induced by salivary gland dysfunction negatively affects the quality of life of patients with diabetes by interfering with their eating and speaking functions and nutritional state, enhancing the risks of oral infection, dental caries, and periodontal diseases [4]. Despite intensive research on the alterations in salivary flow and components in patients with diabetes, the molecular mechanism of diabetic hyposalivation remains largely unknown.

Insulin resistance induced by obesity and high fat diet is one of the major pathogenic factors of type 2 diabetes (T2D). Studies have suggested that the obesity and high fat diet induced insulin resistance contributed to the dysfunction of salivary glands. It has been reported that high fat diet caused obese-insulin resistance, which not only leads to the pathophysiological alterations of salivary glands, including impaired intracellular calcium transients, increased oxidative stress, inflammation, and salivary mitochondrial dysfunction, but also changes the lipid profiles in salivary glands and disrupts salivary redox homeostasis [5,6,7]. Moreover, the corresponding changes in the composition of saliva can reflect the function of salivary gland. As showed in premenopausal women, plasma adiponectin levels increased as a function of salivary pH, suggesting that salivary pH is a significant correlate of plasma adiponectin levels in women [8]. With the increasing prevalence of type 2 diabetes and obesity, saliva may be promising avenue with simple, inexpensive and non-invasive advantages to monitor health status, disease onset, and progression.

Tight junctions (TJs) are cell-cell interactions located in the most apical region between polarized cells [9]. The integrity of TJs is crucial in maintaining paracellular secretion of salivary components [10,11]. A functioning TJ is composed of transmembrane proteins, intracellular proteins, and scaffolding proteins. As major transmembrane components of TJs, 27 claudin proteins have been identified, and these proteins constitute the backbone of TJ strands and characteristically form paracellular barriers and pores to determine paracellular permeability [11]. Knockout of claudin-1 in mice leads to loss of the TJ barrier to water and macromolecules at the stratum granulosum of the epidermis; consequently, these mice die of dehydration in the neonatal period [12]. Claudin-3 is decreased in the submandibular gland (SMG) and positively correlated with salivary function in patients with IgG4-related sialadenitis, a chronic fibroinflammatory disease characterized by salivary gland swelling and hyposalivation [13]. Differential expression patterns of claudins are detected in the salivary glands among species. The human SMG expresses claudin-1 to claudin-12 and claudin-16. However, claudin-6, claudin-9, and claudin-11 are not expressed in the mouse SMG [14]. Claudin-1, claudin-3, claudin-4, claudin-5, and claudin-7 are expressed in the rat SMG [15]. Although several claudins have been demonstrated to play roles in salivary gland diseases, their alterations and underlying mechanism in diabetes-induced salivary gland dysfunction remain to be elucidated.

MicroRNAs (miRNAs) are highly conserved single-stranded noncoding RNAs of approximately 18-24 nucleotides. MiRNAs regulate gene expression by initiating the degradation of target mRNAs or inhibiting mRNA translation, thereby participating in various physiological and pathological processes including diabetes and its complications [16,17]. Recently, miRNAs have been shown to modulate the expression of TJ proteins and regulate epithelial and endothelial barrier functions [18,19]. In a hypoxia-induced brain injury mouse model, miR-212 and miR-132 are upregulated, which impairs blood-brain barrier function by targeting claudin-1, junctional adhesion molecule 3, and TJ-associated protein 1 [20]. Inhibition of miR-155 increases claudin-1 and zonula occludens-1 (ZO-1) expression by directly targeting the claudin-1 mRNA 3′-untranslated region (3′-UTR) and strengthens endothelial TJs after oxygen-glucose deprivation in human primary brain microvascular endothelial cells [21]. Moreover, miR-146b-3p increases claudin-5 and ZO-1 expression in human retinal endothelial cells by inhibiting inflammatory factor secretion in macrophages obtained from patients with diabetic retinopathy, thereby reducing diabetes-induced blood-retinal barrier damage [22]. Therefore, miRNAs may play roles in salivary gland function by regulating TJs.

The present study was designed to investigate the alterations in claudins in the SMG of a type 2 diabetes (T2D) mouse model and explore the regulatory mechanism by which miRNAs affect claudins in the diabetic SMG.

## 2. Material and Methods

### 2.1. Mice

Male db/db mice (T2D group, 8 weeks old) and age-matched male db/m mice (control group), purchased from ChangZhou Cavens Laboratory Animal Ltd. (Changzhou, China), were raised under constant temperature and humidity conditions with free access to water and food for 8 weeks. The animals were fasted for at least 6 h with water available ad libitum before extraction. Blood glucose and serum insulin levels were measured with a glucometer (ACCU-CHEK) and the Iodine [^125^I]-Insulin Radioimmunoassay Kit (Union Medical & Pharmaceutical Technology Ltd., Tianjin, China), respectively. The insulin resistance index (HOMA-IR) was calculated with the formula fasting insulin (mU/L) × fasting glucose (mmol/L)/22.5. After intraperitoneal injection of chloral hydrate (0.4 g·kg^−1^ body weight), the SMGs were extracted from mice, frozen in liquid nitrogen, and then stored at −80 °C. All procedures were approved by the Ethics Committee of Animal Research, Peking University (No. LA2015071) and complied to ARRIVE guidelines.

### 2.2. Cell and Human Submandibular Gland Tissue Culture

The rat SMG acinar cell line SMG-C6 (a kind gift from Dr David O. Quissell) was cultured in DMEM/F12 (1:1 mixture) as previously described [23]. In high-glucose (HG) experiments, cells were incubated in normal medium supplemented with D-glucose at a final concentration of 25 mmol/L. Human SMG tissues were collected from 10 patients who underwent functional neck dissection for oral and maxillofacial malignant tumors without irradiation or chemotherapy at the Department of Oral and Maxillofacial Surgery, Peking University School and Hospital of Stomatology. All SMGs were confirmed to be histologically normal. Detailed information on the patients is listed in Appendix A. The research protocol was approved by the Peking University Institutional Review Board, and was conducted in accordance with the Declaration of Helsinki. All participants signed an informed consent document before tissue collection. Fresh SMGs were transported, cut, and incubated with or without HG medium for the indicated times, as described previously [23].

### 2.3. Transmission Electron Microscopy

SMG tissues were fixed in 2% glutaraldehyde and postfixed in 1% osmium tetroxide. Ultrathin sections were cut and stained with 10% uranyl acetate and 1% lead citrate. The ultrastructure of TJs was observed by using a transmission electron microscope (HITACHI H-7000, Tokyo, Japan), and 5 fields of each sample were randomly chosen for evaluation of TJ width in the SMG by using ImageJ software as described previously [24].

### 2.4. Western Blot Analysis

SMGs and cells were homogenized in RIPA buffer (Thermo Fisher Scientific, Waltham, MA, USA) and centrifuged for 15 min at 4 °C. The supernatant was collected, and the protein concentration was measured by the Bradford method (M&C Gene Technology Ltd., Beijing, China). Equal amounts of proteins (20 μg) were separated by 10% or 12% SDS-PAGE and transferred to polyvinylidene difluoride (PVDF) membranes, which were blocked with 5% nonfat milk, probed with primary antibodies, and then incubated with horseradish peroxidase-conjugated secondary antibodies. Detailed information on the antibodies is listed in Appendix A. Immunoreactive bands were visualized with an enhanced chemiluminescence reagent (Thermo Fisher Scientific, Waltham, MA, USA). Band intensity was calculated with ImageJ software v1.8.0 (National Institutes of Health, Bethesda, MD, USA).

### 2.5. Reverse Transcription PCR and Quantitative Real-Time PCR

Total RNA was extracted and reverse transcribed into cDNA with 5 × All-In-One RT MasterMix (G490, Applied Biological Materials Inc., Richmond, Canada) according to the manufacturer’s instructions. Quantitative real-time PCR (qRT-PCR) was performed on a PikoReal Real-Time PCR System using the DyNAmo™ ColorFlash SYBR Green qPCR Kit (Thermo Fisher Scientific) and analyzed with PikoReal 2.0 software. The primer sequences are shown in Appendix A. MiRNA quantification was determined by the miDETECT A Track qRT-PCR Starter Kit (RiboBio, Guangzhou, China) according to the manufacturer’s instructions. The qRT-PCR primers specific for miR-22-3p and U6 were designed by RiboBio.

### 2.6. Immunofluorescence Staining

Frozen sections (10 μm) of SMGs were immersed in citrate buffer (pH 6.0) and heated in a microwave oven for antigen retrieval. SMG-C6 cells cultured on glass coverslips were fixed in 4% paraformaldehyde and permeabilized with 0.1% Triton X-100. Sections and cells were blocked with 1% bovine serum albumin, incubated with anti-claudin-1, anti-claudin-3, anti-claudin-4 or anti-Sp1 antibodies at 4°C overnight, and then incubated with Alexa Fluor 594- or Alexa Fluor 488-conjugated secondary antibodies. Nuclei were stained with 4′,6-diamidino-2-phenylindole (DAPI). Confocal images were captured by a laser scanning confocal microscope (TCS SP8, Leica, Germany).

### 2.7. Knockdown of Claudin-1, Claudin-3 and Sp1

GFP-tagged shRNA constructs specific for rat claudin-1 and claudin-3 and a scramble control were constructed with pGFP-V-RS vectors (Origene, MD, USA). The Sp1-specific siRNA sequence, nonspecific negative control, and positive control were synthesized by HanBio company. The sequences are listed in Appendix A. SMG-C6 cells were cultured to 60% confluency and transfected with the shRNA or siRNA of interest by using Lipofectamine^®^ 2000 (Thermo Fisher Scientific) according to the manufacturer’s instructions. Claudin-1- and claudin-3-knockdown cells were selected with puromycin (10 μg/mL, HanBio, Shanghai, China). The stably transfected clones were isolated and passaged.

### 2.8. Measurement of Transepithelial Electrical Resistance and Paracellular Tracer Flux

A confluent monolayer of SMG-C6 cells was grown in a 24-well Corning Transwell™ filter (6.5-mm diameter, 0.4-μm pore size). Transepithelial electrical resistance (TER) was measured using an Epithelial Volt Ohm Meter EVOM2 (World Precision Instruments, Sarasota, FL, USA). All TER values were obtained from at least three wells. Final values were calculated by subtracting the blank filter value (80 Ω) and multiplying by the surface area of the filter. For the paracellular tracer flux assay, 1 mg/mL 4-kDa (68059, Sigma-Aldrich, St. Louis, MO, USA) or 40-kDa FITC-dextran (53379, Sigma-Aldrich) was added to the medium in the basal compartment, the medium in the apical compartment was collected after an incubation at 37 °C for 3 h, and the paracellular tracer flux was measured using EnSpire Multilabel Plate Reader (PerkinElmer, Waltham, MA, USA).

### 2.9. Plasmid Construction and a Dual-Luciferase Activity Assay

TargetScan database was used to predict the potential binding sites for miR-22-3p in the Sp1 mRNA 3′-UTR. The sequence of the Sp1 mRNA 3′-UTR containing the binding site or a mutant sequence was cloned into the pGL3 vector (Promega, Madison, WI, USA), and a luciferase reporter plasmid was constructed. SMG-C6 cells were cotransfected with a luciferase reporter plasmid and miR-22-3p mimic or inhibitor (RiboBio). Luciferase activity was measured using a dual-luciferase reporter assay system according to the manufacturer’s protocol (Promega) and determined with a luminometer (BioTek, Winooski, VT, USA). Relative luciferase activity was calculated with normalization to the Renilla luciferase activity.

### 2.10. Chromatin Immunoprecipitation Assay

A chromatin immunoprecipitation (ChIP) assay was performed with a ChIP kit (#9005, Cell Signaling Technology, Danvers, MA, USA). Cells cultured in 15-cm plates were crosslinked with 1% formaldehyde, quenched with 0.125 mmol/L glycine, and then lysed. DNA was fragmented using an ultrasonicator (SONICS, Tokyo, Japan). Protein-DNA complexes were immunoprecipitated with anti-Sp1 or IgG antibodies at 4 °C overnight. The crosslinked complexes were washed and decrosslinked at 65 °C for 30 min. The pulled-down DNA fragments and input DNA were purified and used for qRT-PCR analysis with primers designed to amplify the claudin-1 and claudin-3 promoter regions containing the putative Sp1 binding sites. The primer sequences are listed in Appendix A.

### 2.11. Transfection of a MiR-22-3p Mimic or Inhibitor

A miR-22-3p mimic, a miR-22-3p inhibitor, and the corresponding negative controls (NCs) were purchased from RiboBio. SMG-C6 cells were cultured until 50% confluent and then transiently transfected with the miR-22-3p mimic, the miR-22-3p inhibitor, or a NC by using the ribo FECT™ CP Transfection Kit (C10511, RiboBio) according to the manufacturer’s instructions. After 24 h, the cells were treated with or without HG medium.

### 2.12. Statistical Analysis

Data are presented as the mean ± standard error of the mean (SEM). Statistical analysis was carried out using an unpaired Student’s *t*-test for two-group comparisons. Multiple groups were compared with one-way or two-way ANOVA followed by Bonferroni’s test with GraphPad Prism 5 (GraphPad Software, San Diego, CA, USA). *p* < 0.05 was considered to be statistically significant.

## 3. Results

### 3.1. Expression of Claudin-1 and Claudin-3 Is Upregulated in the Diabetic Submandibular Gland

The blood glucose and serum insulin levels and HOMA-IR of db/db mice were higher than those of db/m mice, confirming the establishment of a T2D mouse model (Appendix A). Transmission electron microscopy images of SMGs showed that the TJs appeared as a narrow and continuous seal in the apical region between neighboring acini in db/m mice, while the lateral membranes of tight junctions appeared to be less well defined but closely aligned compared with db/m mice (Figure 1A,B). Quantitative analysis showed that the average TJ width, an indicator of TJ opening, was decreased in the SMGs of db/db mice (Figure 1C).

To identify the alterations in TJ components in the diabetic SMG, the expression and distribution of claudins were detected. The mRNA and protein levels of claudin-1 and claudin-3 were significantly increased, while the claudin-4 mRNA and protein levels were decreased in the SMGs of db/db mice (Figure 1D–G). No differences were observed in claudin-2, claudin-5, claudin-7 or claudin-10 expression between db/db mice and db/m mice (Figure 1D,H,I). Immunofluorescence staining images showed that claudin-1 and claudin-3 were predominantly expressed in the apicolateral and basolateral membranes of acini in the mouse SMG, and their staining intensities were enhanced in the db/db group. Claudin-4 was mainly located in the apicolateral membrane of ducts, and consistent with Western blot analysis results, the fluorescence intensity of claudin-4 was reduced in the diabetic SMG. However, no obvious differences were observed in the locations of claudin-1, claudin-3, and claudin-4 between the db/db and db/m groups (Figure 1J).

### 3.2. High Glucose Increases the Expression of Claudin-1 and Claudin-3 in SMG-C6 Cells and Cultured Human Submandibular Gland Tissues

To explore the role of HG in claudin expression, SMG-C6 cells were incubated in 25 mmol/L glucose for the indicated times. The amounts of claudin-1 and claudin-3 were significantly increased, whereas the expression of claudin-4 was unaffected (Figure 2A–C). Mannitol (25 mmol/L), used as an osmotic pressure control, did not affect the expression of claudin-1 or claudin-3 (Figure 2D–G). Immunofluorescent images and quantitative analysis showed higher fluorescent intensities of claudin-1 and claudin-3 at 12 and 24 h in HG group, but claudin-4 did not change. However, this is not occurred in high mannitol (HM) group (Appendix A). Moreover, elevated claudin-1 and claudin-3 expression was observed at both the mRNA and protein levels in fresh cultured human SMG tissue after HG treatment for 24 h, whereas the expression of claudin-4 and claudin-7 was unchanged (Figure 2H–M). These results indicate that HG can directly upregulate claudin-1 and claudin-3 expression.

### 3.3. High Glucose Decreases the Paracellular Permeability of SMG-C6 Cells

To define the effect of HG on TJ barrier function, TER values were measured to analyze the paracellular permeability of SMG-C6 cells. The basal TER value was consistent with that in a previous report on the same cell line [25], while HG treatment of SMG-C6 cells increased the TER value starting at 12 h and elicited the greatest increase at 24 h (Figure 3A). Paracellular permeability was also evaluated by using FITC-dextran as a noncharged paracellular tracer. The 4-kDa FITC-dextran flux dropped at 12 and 24 h after HG treatment (Figure 3B). Meanwhile, HG markedly reduced the paracellular flux of 40-kDa FITC-dextran starting at the first hour (Figure 3C). These results indicate that HG decreases the paracellular permeability of SMG-C6 cells.

### 3.4. Claudin-1 and Claudin-3 Are Required for the High Glucose-Induced Reduction in Paracellular Permeability

To further explore whether the increased claudin-1 and claudin-3 expression was related to the reduction in paracellular permeability in HG-cultured SMG-C6 cells, we established SMG-C6 cells with stable knockdown of claudin-1 or claudin-3 by shRNA transfection. The expression of claudin-1 was significantly reduced in claudin-1-knockdown cells, while the expression of claudin-3 and claudin-4 was unaffected (Figure 3D,E). A significant decrease in claudin-3 expression, but not claudin-1 or claudin-4 expression, was present in the claudin-3-knockdown group (Figure 3F,G). The basal TER values were significantly decreased, whereas the 4-kDa and 40-kDa FITC-dextran fluxes were markedly increased in both claudin-1- and claudin-3-knockdown SMG-C6 cells. Moreover, the HG-induced higher TER values and lower FITC-dextran fluxes were abolished (Figure 3H–J), suggesting that claudin-1 and claudin-3 play crucial roles in epithelial barrier formation in SMG-C6 cells and mediate the HG-modulated alteration in paracellular permeability.

### 3.5. MiR-22-3p Is Downregulated and Involved in the High Glucose-Induced Upregulation of Claudin-1 and Claudin-3

In our previous study, we identified 28 differentially expressed miRNAs in the SMGs of db/db mice, and the bioinformatic analysis of these miRNAs revealed an enrichment in the TJ signaling pathway, suggesting that the miRNAs might be involved in the regulation of TJs in the diabetic SMG [26]. The raw high-throughput sequencing data have been made publicly available at the Gene Expression Omnibus (GEO) database under the number GSE141412. The top 3 upregulated miRNAs (miR-181a-5p, miR-375-3p, and miR-99b-5p) and downregulated miRNAs (miR-145a-5p, miR-143-3p, and miR-22-3p) in the diabetic SMG were assessed by qRT-PCR (Figure 4A). The expression of miR-375-3p, miR-99b-3p, miR-143-3p, and miR-22-3p was consistent with that detected with high-throughput technology (Figure 4B). Furthermore, miR-375-3p was upregulated, while miR-143-3p and miR-22-3p were downregulated in HG-cultured SMG-C6 cells compared with control cells (Figure 4C). Among these miRNAs, miR-22-3p was the most significantly downregulated. In cultured human SMGs, HG also reduced miR-22-3p expression (Figure 4D).

MiR-22-3p has been demonstrated to regulate lipid, insulin, and glucose metabolism and play a crucial role in the development of metabolic disorders such as obesity, diabetes and diabetic complications [27,28,29]. To classify the role of miR-22-3p in the diabetic SMG, we incubated SMG-C6 cells with HG medium for the indicated times and found that miR-22-3p expression was attenuated (Figure 4E). A miR-22-3p mimic or inhibitor was transfected into SMG-C6 cells 24 h before an incubation in HG medium. The efficiencies of the miR-22-3p mimic and inhibitor were confirmed by qRT-PCR (Figure 4F,G). Overexpression of miR-22-3p with the mimic (100 nmol/L) effectively inhibited claudin-1 and claudin-3 expression and reversed the HG-induced upregulation of claudin-1 and claudin-3 (Figure 4H). In contrast, the expression of claudin-1 and claudin-3 was evidently increased by transfecting the miR-22-3p inhibitor (100 nmol/L) (Figure 4I). Our findings indicate that miR-22-3p negatively regulates claudin-1 and claudin-3 expression. The reduction in the miR-22-3p level contributes to the HG-induced upregulation of claudin-1 and claudin-3.

### 3.6. Sp1 Is Increased in the Diabetic Submandibular Gland and High Glucose-Treated SMG-C6 Cells

To identify the mechanism by which miR-22-3p negatively regulates claudin-1 and claudin-3 expression, we predicted miR-22-3p target genes with the TargetScan database and found that Sp1, a transcription factor of claudins, is a direct target of miR-22-3p. The mRNA and protein levels of Sp1 were higher in the SMGs of db/db mice than in those of db/m mice (Figure 5A,B). Sp1 was distributed in both the cytoplasm and the nucleus in SMG tissues from db/m mice, whereas significantly enhanced Sp1 immunostaining was observed in SMG tissues from db/db mice, especially in the nucleus (Figure 5C). Consistently, Sp1 was strongly upregulated in SMG-C6 cells after 6 h of exposure to HG medium (Figure 5D,E). Immunofluorescence staining for Sp1 was predominantly found in the nucleus of SMG-C6 cells, and the nuclear staining intensity was much stronger in the HG group than in the control group (Figure 5F). In cultured human SMGs, Sp1 expression was increased after exposure to HG for 24 h (Figure 5G–I).

### 3.7. Sp1 Activates the Transcription of Claudin-1 and Claudin-3 by Binding to the Claudin-1 and Claudin-3 Promoters

Bioinformatic prediction with the JASPAR and ALGEN databases showed that several Sp1 binding sites existed in the promoter of claudin-1 spanning from −400 to 315 bp and that of claudin-3 spanning from −1677 to 396 bp. Therefore, we evaluated the claudin-1 and claudin-3 promoter regions in several fragments, with each fragment being approximately 200 bp. ChIP assays were performed to verify whether Sp1 binds to the promoter regions of claudin-1 and claudin-3. The interaction between Sp1 and a claudin-1 promoter motif in the −284 to −84 bp region was obviously increased under HG conditions (Figure 6A). Furthermore, HG promoted interactions between Sp1 and the claudin-3 promoter in 3 specific motifs within the following regions: −1677 to −1480 bp, −364 to −146 bp, and −147 to 52 bp (Figure 6B).

To determine the precise regulatory role of Sp1 in claudin-1 and claudin-3 expression, we inhibited Sp1 by using the specific Sp1 inhibitor mithramycin A (mitA) or siRNA. MitA (100 nmol/L) significantly inhibited Sp1, claudin-1, and claudin-3 expression in SMG-C6 cells (Figure 6C–F). Preincubation with 100 nmol/L mitA not only reduced claudin-1 and claudin-3 expression but also suppressed their HG-induced upregulation (Figure 6G–J). The efficiency of the Sp1-specific siRNA was confirmed by Western blot analysis (Figure 6K,L). Silencing endogenous Sp1 repressed the expression of claudin-1 and claudin-3. Moreover, the effects of HG on claudin-1 and claudin-3 disappeared with Sp1-specific siRNA treatment (Figure 6M–P).

### 3.8. MiR-22-3p Upregulates Claudin-1 and Claudin-3 by Directly Targeting Sp1 in SMG-C6 Cells

Bioinformatic prediction using TargetScan indicated that the Sp1 mRNA 3′-UTR contains one conserved binding site for miR-22-3p (Figure 7A). To confirm that Sp1 is a direct target of miR-22-3p, luciferase reporter plasmids containing the wild-type Sp1 3′-UTR segment or a mutated sequence were constructed (Figure 7B). A luciferase reporter assay showed that the miR-22-3p mimic reduced the luciferase activities of SMG-C6 cells transfected with the Luc-Sp1-3′-UTR-Wt plasmid; conversely, luciferase activity was increased by the miR-22-3p inhibitor. However, the luciferase activity was unaffected by cotransfection of either the miR-22-3p mimic or the miR-22-3p inhibitor in the Luc-Sp1-3′-UTR-Mut group (Figure 7C). Moreover, HG treatment enhanced luciferase activity in the Luc-Sp1-3′-UTR-Wt group but had no effect in either the vector or Luc-Sp1-3′-UTR-Mut group (Figure 7D). These results demonstrate that Sp1 is a direct target of miR-22-3p and that HG attenuates the direct interaction between miR-22-3p and Sp1. Furthermore, overexpression of miR-22-3p apparently reduced Sp1 expression and reversed the HG-induced upregulation of Sp1 (Figure 7E). Downregulation of miR-22-3p conversely increased Sp1 expression in SMG-C6 cells (Figure 7F).

## 4. Discussion

In the present study, we demonstrated the important role of the miR-22-3p/Sp1 pathway in modulating claudin-1 and claudin-3 expression in the SMG. In the diabetic SMG, miR-22-3p was markedly downregulated, which increased the expression of the transcription factor Sp1 by suppressing the interaction between miR-22-3p and the Sp1 3′-UTR. The enhanced Sp1 level activated claudin-1 and claudin-3 transcription by binding directly to the promoter regions of claudin-1 and claudin-3, thereby reducing the paracellular permeability of SMG acinar cells (Figure 7G). These findings provide new insights into the important role of miR-22-3p in SMG function and reveal a new mechanism of diabetic hyposalivation.

The maintenance of TJ function has been shown to play critical roles in both water secretion and secretory granule exocytosis in the process of salivary secretion [30]. Alterations in the molecular composition of TJs affect the paracellular permeability of the salivary gland epithelium and are involved in pathological processes, such as Sjögren’s syndrome and radiation injury [31,32]. Among the multiple molecules composing TJs, claudins are indispensable in forming epithelial barriers and transportation systems [33]. A previous study reported that activation of muscarinic acetylcholine receptors regulates the paracellular transport of SMG epithelial cells by reducing claudin-4 expression [25]. In salivary inflammatory diseases, tumor necrosis factor-α increases the paracellular permeability of SMG-C6 cells by inducing the loss of claudin-3 in the TJ structure [34]. TJ function and integrity are influenced by obesity associated diabetes. Obese and diabetic mice model ob/ob mice display enhanced intestinal permeability by reducing ZO-1 and occludin expression, which participates in the occurrence of intestinal disorders [35]. The breakdown of blood-brain barrier caused by eroding tight junctions is observed in obesity and high fat diet mice [36]. Here, the expression of claudin-1 and claudin-3 was significantly increased in the SMGs of diabetic mice. Hyperglycemia is a main pathogenic factor of diabetes, and HG incubation increased claudin-1 and claudin-3 expression and reduced paracellular permeability in SMG-C6 cells. Knockdown of claudin-1 or claudin-3 increased cell paracellular permeability and abolished the effect of HG on paracellular permeability. These results indicate that claudin-1 and claudin-3 play vital roles in modulating TJ function in SMG acinar cells and that upregulated claudin-1 and claudin-3 mediate the HG-induced decrease in the paracellular permeability of SMG epithelial cells under diabetic conditions. Although the expression of claudin-4 was reduced in the SMGs of T2D mice, HG did not induce changes in SMG cells or the human SMG. Our previous study showed that activation of AMP-activated protein kinase induced a redistribution of claudin-4 but not claudin-1 or claudin-3 in submandibular epithelial cells [37]. The results suggested that the expression of claudin-4 might be regulated by a different mechanism. Interestingly, the effects of HG on SMG epithelium may be different with other tissues such as intestinal epithelium. HG treated human intestinal Caco-2 cells present decreased TER and increased paracellular permeability, caused by reduced expression of claudin-1, occludin, and ZO-1 [38,39]. Moreover, by regulating Na/K-ATPase, HG enhances the absorption of glucose and NaCl in diabetic intestine [40]. HG induced higher intestinal permeability promotes infections and leads to intestinal disorders [41]. The inconsistent results between salivary and intestinal epithelium revealed that TJ in different cell types may play distinct functions.

The function of TJs is dynamically modulated by various regulatory mechanisms. MiRNAs have recently been suggested to regulate TJ proteins and modulate epithelial and endothelial barrier functions in different tissues, such as intestinal epithelium, renal epithelium, blood-brain barrier, and blood-retinal barrier [42,43,44]. MiR-23a attenuates ZO-1 expression through posttranscriptional binding to the ZO-1 3′-UTR in lung epithelial cells and is involved in HIV-mediated lung epithelial barrier dysfunction [45]. MiR-21 is reported to increase the expression of occludin by targeting Rho-associated protein kinase 1 and plays a protective role in the prevention of intestinal barrier dysfunction [46]. Moreover, claudin-3 expression is indirectly affected by miR-93 through modulation of protein tyrosine kinase 6, which mediates intestinal epithelial dysfunction during inflammatory injury [47]. Thus, miRNAs can regulate TJ function in a direct or indirect manner [48]. Based on the differentially expressed miRNAs in the SMGs of T2D model mice, we found that miR-22-3p was markedly downregulated in the diabetic SMG and HG-treated SMG cells. MiR-22-3p is reported to be downregulated in the islet tissues of mice with gestational diabetes, and overexpression of miR-22-3p efficiently improves fasting blood glucose levels and insulin resistance in mice [29]. The long noncoding RNA MIAT is increased in a rat model of diabetic cardiomyopathy and HG-cultured neonatal cardiomyocytes, which upregulates death-associated protein kinase 2 expression by sponging miR-22-3p and consequently leads to cardiomyocyte apoptosis [28]. Moreover, Mir22-knockout mice present dramatically exacerbated fat mass gain, hepatomegaly, and a liver steatosis response to high-fat diet feeding compared to control mice [49]. Although miR-22-3p has been demonstrated to be a master regulator of lipid and glucose metabolism with differential effects on specific organs, its roles in the physiological and pathological processes of the salivary gland remain unknown. Here, the miR-22-3p mimic reduced claudin-1 and claudin-3 expression and diminished the HG-induced upregulation of claudin-1 and claudin-3, whereas the miR-22-3p inhibitor increased claudin-1 and claudin-3 expression. These results demonstrate that miR-22-3p is a negative regulator of claudin-1 and claudin-3 expression in SMG cells.

The main function of miRNAs is to regulate posttranscriptional gene expression by binding to the 3′-UTR of target mRNAs, causing translational suppression or gene activation. Bioinformatic analysis predicted that the transcription factor Sp1, but not claudin-1 or claudin-3, contains miR-22-3p binding sites. Herein, a dual-luciferase reporter assay showed that Sp1 was a specific target of miR-22-3p and that HG reduced the interaction between miR-22-3p and Sp1. The miR-22-3p mimic repressed Sp1 expression and the HG-induced upregulation of Sp1, whereas the miR-22-3p inhibitor enhanced these events, indicating that miR-22-3p negatively regulates Sp1 expression by binding to the Sp1 3′-UTR.

Several studies have shown that the claudin-1 and claudin-3 promoter regions contain Sp1 binding sites and that mutations in these regions result in a significant loss of claudin-1 and claudin-3 transcription [50]. In this study, Sp1 was upregulated and predominantly located in the nucleus in the diabetic SMG and HG-treated SMG cells. ChIP assays verified that HG increased the binding of Sp1 with the claudin-1 promoter in the region −284 to −84 bp in SMG-C6 cells, in which the binding site for Sp1 in the claudin-1 promoter was similar to others in intestinal epithelial cells and human breast adenocarcinoma cell line MCF-7 (−133 to −61 bp and −138 to −76 bp) [51,52]. Moreover, the binding between Sp1 and motifs in the claudin-3 promoter regions −1677 to −1480 bp, −364 to −146 bp, and −147 to 52 bp was enhanced under HG conditions. Sp1 is reported to bind to the claudin-3 promoter region at −112 to −74 bp in ovarian cancer cells, which is consistent with our result of −147 to 52 bp [53]. However, two new binding regions spanning from −1677 to −1480 bp and −364 to −146 bp were found in our study, indicating that the interaction sites of Sp1 involved in regulating claudins may be inconsistent among different cells. Sp1 is normally described as a transcriptional activator, but it can also act as a transcriptional repressor [54,55]. To further identify the precise regulatory roles of Sp1 in claudin-1 and claudin-3 expression in the SMG, a Sp1-specific inhibitor or siRNA was administered. Knockdown of Sp1 relatively suppressed claudin-1 and claudin-3 expression and abolished the higher claudin-1 and claudin-3 levels induced by HG. Our results demonstrate that HG-induced Sp1 expression activates claudin-1 and claudin-3 transcription via direct binding of Sp1 to the promoters of claudin-1 and claudin-3.

## 5. Conclusions

In summary, our study demonstrated an important role for miR-22-3p in regulating SMG epithelial paracellular permeability. During diabetes, inhibition of miR-22-3p increases claudin-1 and -3 expression by directly targeting Sp1, resulting in lower epithelial paracellular permeability in SMG epithelial cells, which might contribute to the dysfunction of the diabetic SMG. These findings extend our understanding of the classic miRNA function involved in salivary secretion and reveal a novel mechanism of diabetic hyposalivation, which might provide an experimental basis for therapeutic strategies for alleviating the oral health issues of patients with diabetes.

## Figures and Tables

**Figure 1 cells-10-03230-f001:**
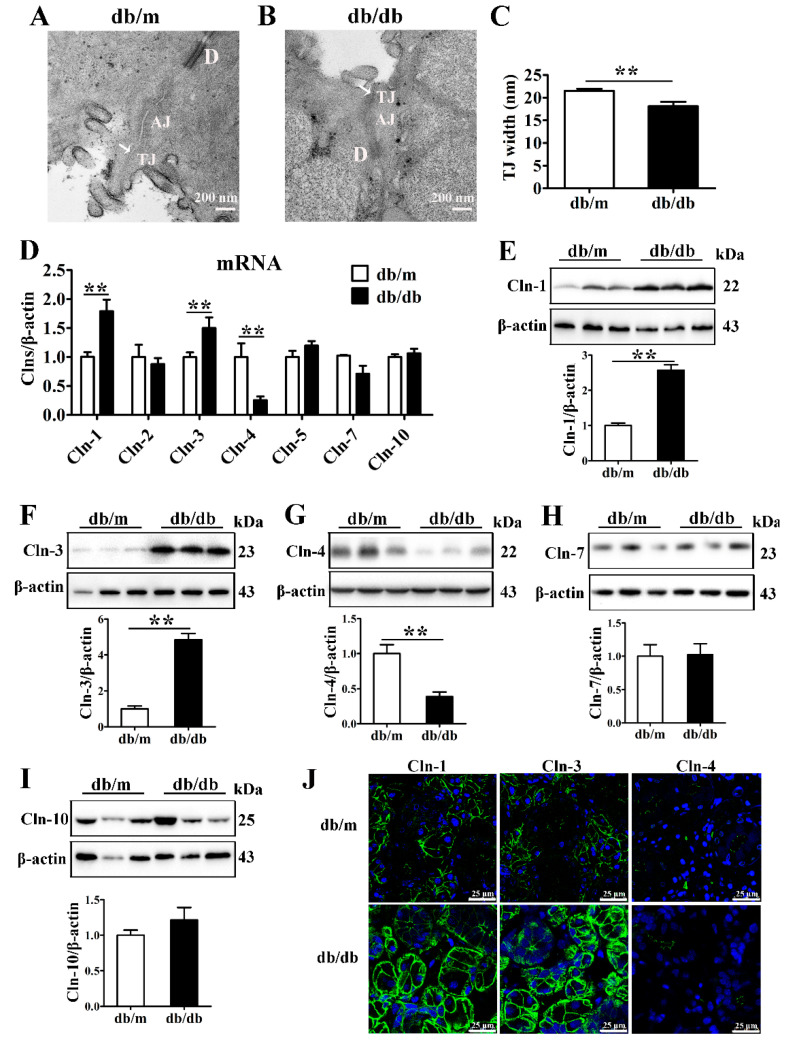
Expression of claudin-1 and claudin-3 is upregulated in the diabetic submandibular gland. (**A**,**B**) Transmission electron microscopy images of tight junctions (TJs, white arrows) in the submandibular glands (SMGs) of db/m mice (**A**) and db/db mice (**B**). (**C**) Statistical analysis of TJ width in mouse SMGs. *n* = 6. (**D**) QRT-PCR analysis of the mRNA levels of claudins (Clns) in mouse SMGs. *n* = 8. (**E**–**I**) Western blot and densitometry analysis of Cln-1 (**E**), Cln-3 (**F**), Cln-4 (**G**), Cln-7 (**H**), and Cln-10 (**I**) in mouse SMGs. *n* = 6–8. (**J**) Immunofluorescence staining for Cln-1, Cln-3 and Cln-4 in mouse SMGs. ** *p* < 0.01. AJ, adherens junction; D, desmosome.

**Figure 2 cells-10-03230-f002:**
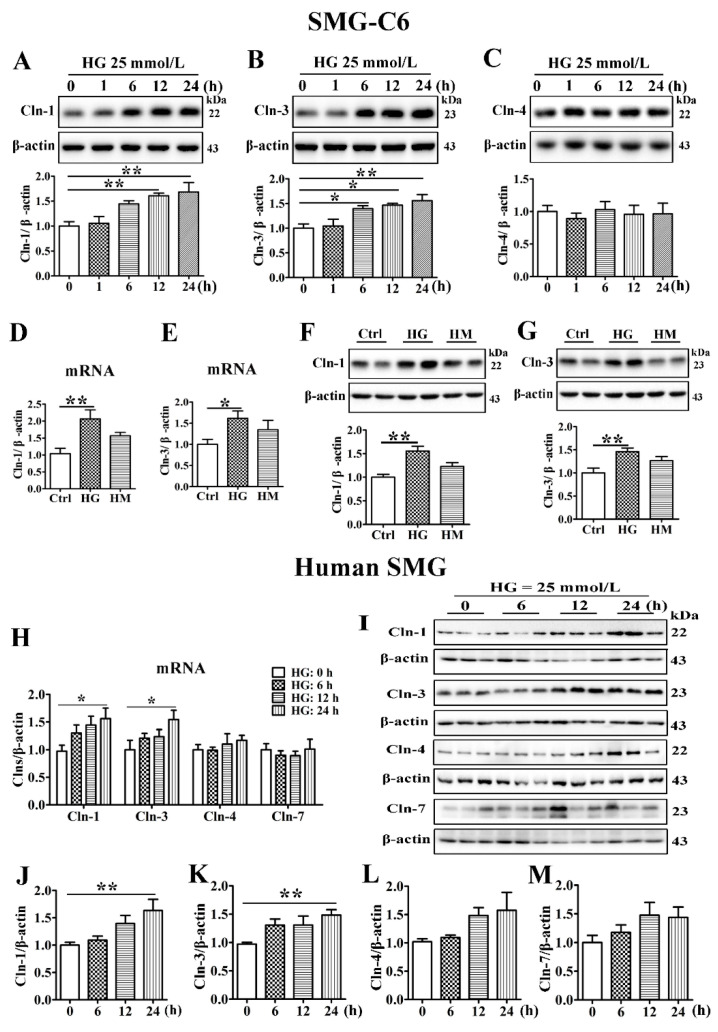
High glucose increases the expression of claudin-1 and claudin-3 in SMG-C6 cells and cultured human submandibular gland tissues. (**A**–**C**) Western blot and densitometry analysis of claudin-1 (Cln-1) (**A**), Cln-3 (**B**), and Cln-4 (**C**) expression in high glucose (HG = 25 mmol/L)-treated SMG-C6 cells. *n* = 6. (**D**–**G**) QRT-PCR and western blot analysis of Cln-1 and Cln-3 expression in the control (Ctrl), HG, and high mannitol (HM = 25 mmol/L) groups. *n* = 6. (**H**) QRT-PCR analysis of the mRNA levels of Clns in HG-treated human submandibular glands (SMGs). (**I**–**M**) Western blot (**I**) and densitometry analysis of Cln-1 (**J**), Cln-3 (**K**), Cln-4 (**L**), and Cln-7 (**M**) in HG-treated human SMGs. *n* = 10. * *p* < 0.05 and ** *p* < 0.01.

**Figure 3 cells-10-03230-f003:**
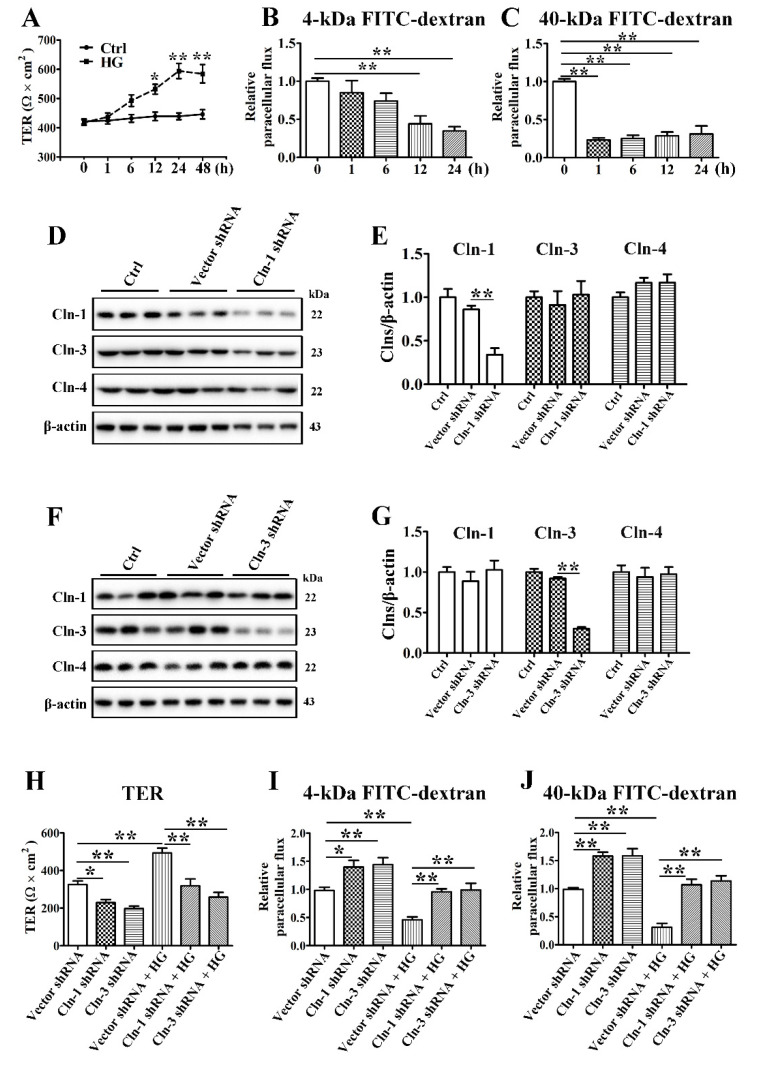
Claudin-1 and claudin-3 are required for a high glucose-induced reduction in paracellular permeability. (**A**) The effect of high glucose (HG = 25 mmol/L) on the transepithelial electrical resistance (TER) of SMG-C6 cells. *n* = 8. (**B**,**C**) The effect of HG on the paracellular flux of 4-kDa FITC-dextran (**B**) and 40-kDa FITC-dextran (**C**). *n* = 6. (**D**,**E**) Western blot (**D**) and densitometry analysis (**E**) of claudin-1 (Cln-1), Cln-3, and Cln-4 after transfecting SMG-C6 cells with Cln-1-specific shRNA. *n* = 3. (**F**,**G**) Western blot (**F**) and densitometry analysis (**G**) of Cln-1, Cln-3, and Cln-4 after transfecting SMG-C6 cells with Cln-3-specific shRNA. *n* = 3. (**H**) TER of Cln-1-knockdown cells and Cln-3-knockdown cells. *n* = 6. (**I**,**J**) Paracellular flux of 4-kDa FITC-dextran (**I**) and 40-kDa FITC-dextran (**J**) in Cln-1-knockdown cells and Cln-3-knockdown cells. *n* = 6. * *p* < 0.05 and ** *p* < 0.01.

**Figure 4 cells-10-03230-f004:**
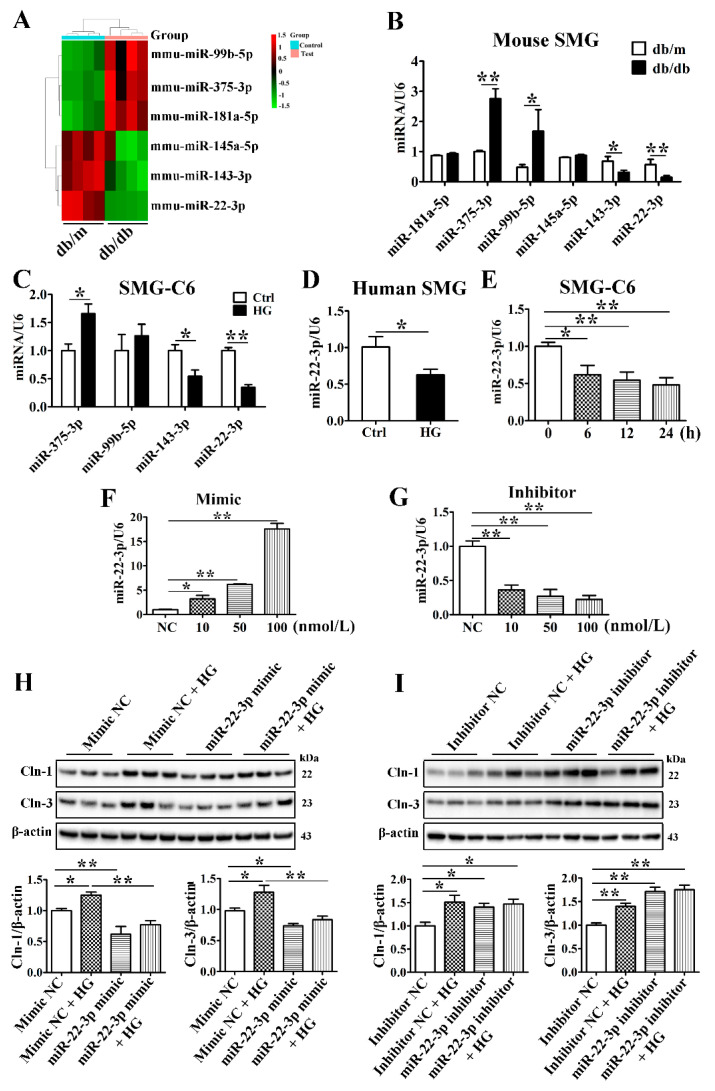
MiR-22-3p is downregulated and involved in the high glucose-induced upregulation of claudin-1 and claudin-3. (**A**) Heat map of the top 3 upregulated and downregulated microRNAs (miRNAs) in the submandibular glands (SMGs) of db/db mice. *n* = 4. (**B**) QRT-PCR analysis of the top 3 upregulated and downregulated miRNAs in the SMGs of db/db mice. *n* = 8. (**C**) QRT-PCR analysis of candidate miRNAs in high glucose (HG = 25 mmol/L)-treated SMG-C6 cells. *n* = 6. (**D**) QRT-PCR analysis of miR-22-3p expression in HG-treated human SMGs. *n* = 10. (**E**) Time course of miR-22-3p expression induced by HG in SMG-C6 cells. *n* = 6. (**F**,**G**) QRT-PCR analysis of the efficiencies of a miR-22-3p mimic and inhibitor. *n* = 3. (**H**,**I**) Western blot and densitometry analysis of Cln-1 and Cln-3 after transfecting SMG-C6 cells with the miR-22-3p mimic (**H**) or inhibitor (**I**). *n* = 6. * *p* < 0.05 and ** *p* < 0.01. NC, negative control.

**Figure 5 cells-10-03230-f005:**
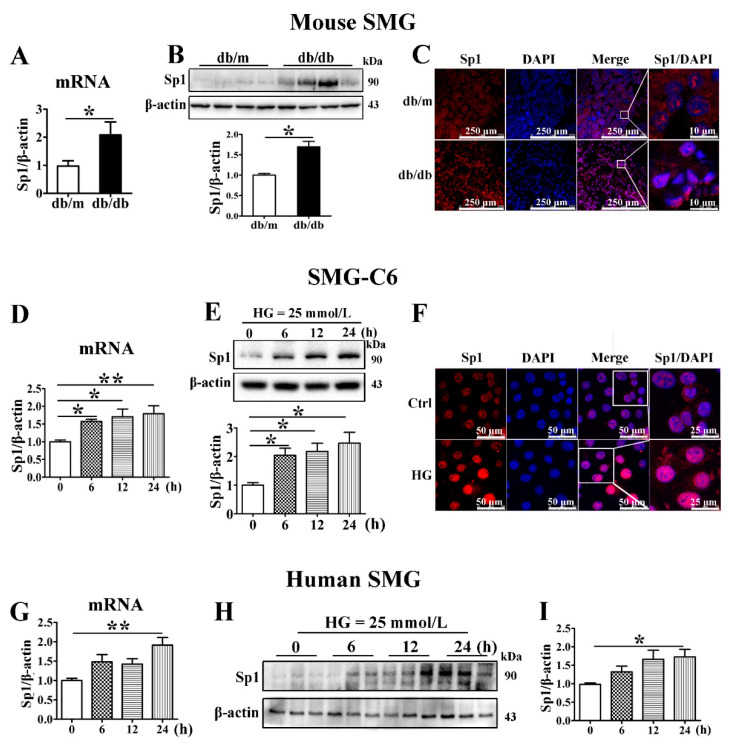
Sp1 is increased in the diabetic submandibular gland and high glucose-treated SMG-C6 cells. (**A**) QRT-PCR analysis of specificity protein-1 (Sp1) mRNA levels in mouse submandibular glands (SMGs). *n* = 8. (**B**) Western blot and densitometry analysis of Sp1 expression in mouse SMGs. *n* = 8. (**C**) Immunofluorescence images showing the distribution and expression of Sp1 in mouse SMGs. (**D**,**E**) Time course of the Sp1 mRNA and protein expression induced by high glucose (HG = 25 mmol/L) in SMG-C6 cells. *n* = 6. (**F**) Immunofluorescence images showing the distribution and expression of Sp1 in HG-treated SMG-C6 cells. (**G**–**I**) Time course of the Sp1 mRNA and protein expression induced by HG (25 mmol/L) in human SMGs. *n* = 10. * *p* < 0.05 and ** *p* < 0.01.

**Figure 6 cells-10-03230-f006:**
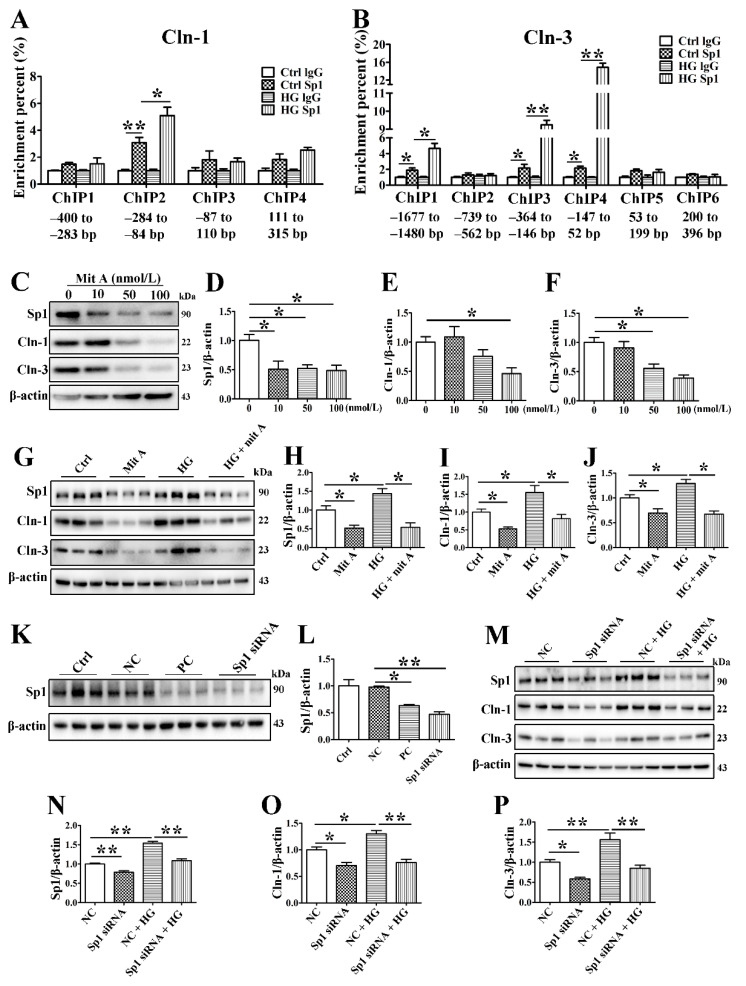
Sp1 activates the transcription of claudin-1 and claudin-3 by binding to the claudin-1 and claudin-3 promoters. (**A**) Chromatin immunoprecipitation (ChIP) assay for detection of the binding between specificity protein-1 (Sp1) and the claudin-1 (Cln-1) promoter. *n* = 6. (**B**) ChIP assay for detecting the binding between Sp1 and the Cln-3 promoter. *n* = 6. (**C**–**F**) Western blot (**C**) and densitometry analysis of the efficiency of the Sp1 inhibitor mithramycin A (mitA). *n* = 3. (**G**–**J**) Western blot (**G**) and densitometry analysis of Sp1 (**H**), Cln-1 (**I**), and Cln-3 (**J**) after incubating SMG-C6 cells with 100 nmol/L mitA. *n* = 6. (**K**,**L**) Western blot and densitometry analysis of the efficiency of Sp1-specific siRNA. *n* = 3. (**M**–**P**) Western blot (**M**) and densitometry analysis of Sp1 (**N**), Cln-1 (**O**), and Cln-3 (**P**) after transfecting SMG-C6 cells with Sp1-specific siRNA. *n* = 6. * *p* < 0.05 and ** *p* < 0.01. Ctrl, control; NC, negative control; PC, positive control.

**Figure 7 cells-10-03230-f007:**
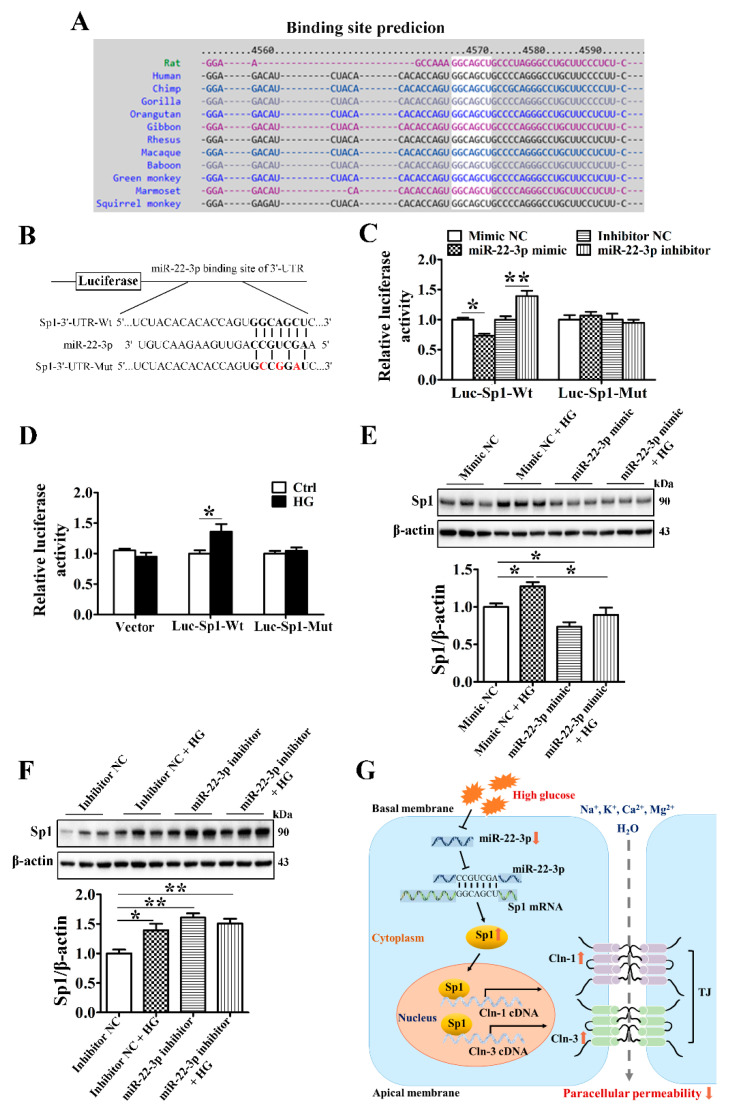
MiR-22-3p upregulates claudin-1 and claudin-3 by directly targeting Sp1 in SMG-C6 cells. (**A**) Bioinformatic prediction using the TargetScan database indicated that the Sp1 mRNA 3′-UTR contained one conserved binding site for miR-22-3p. (**B**) Construction of luciferase reporter plasmids containing the Sp1 wild-type cDNA sequence (Luc-Sp1-3′-UTR-Wt) or a mutant sequence (Luc-Sp1-3′-UTR-Mut). (**C**) Dual-luciferase reporter assay with SMG-C6 cells cotransfected with a luciferase reporter plasmid and miR-22-3p mimic or inhibitor. *n* = 6. (**D**) Dual-luciferase reporter assay with high glucose (HG = 25 mmol/L)-treated SMG-C6 cells transfected with a luciferase reporter plasmid. *n* = 6. (**E**,**F**) Western blot and densitometry analysis of Sp1 after transfecting SMG-C6 cells with the miR-22-3p mimic (**E**) or inhibitor (**F**). *n* = 6. (**G**) Schematic illustration showing the mechanism of the miR-22-3p-mediated reduction in epithelial paracellular permeability in the diabetic SMG. * *p* < 0.05 and ** *p* < 0.01.

## Data Availability

The data presented in this study are available on request from the corresponding authors.

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
