# Peer review of "High Glucose Reduces the Paracellular Permeability of the Submandibular Gland Epithelium via the MiR-22-3p/Sp1/Claudin Pathway"

_cells, 2021, doi:10.3390/cells10113230_

Round 1
Reviewer 1 Report
The work presented in this manuscript is interesting and novel. The manuscript was well-written and is good for Cells: There are two minor concerns:
1) The quality of Figure 1J should be improved
2) There are no scale bars for Figure 5C and F.
Author Response
1) The quality of Figure 1J should be improved
Answer: We appreciate Reviewer #1 for your valuable suggestion. In order to improve the quality of Figure 1J, we selected immunofluorescence images of claudin-1, claudin-3, and claudin-4 with higher quality to replace the images used before.
2) There are no scale bars for Figure 5C and F
Answer: As Reviewer #1 pointed out, we added the scale bars in Figure 5C and F.
Reviewer 2 Report
In this study, the authors show the role of miR-22-3p in regulating SMG epithelial paracellular permeability in a very well designed, executed, and discussed paper, with several sets of experiments and adequate methodologies.
Congratulations to the authors.
Author Response
Thank reviewer 2 for the comments.
Reviewer 3 Report
The authors present an interesting finding in terms of the transcriptional upregulation of claudin-1 and claudin-3 in the presence of high glucose and deduce that perhaps the reduced permeability contributes to hyposalivation in diabetic models. Strengths include analysis of claudins at various timepoints after glucose treatment, both at protein and mRNA levels, and further pointing to a specific MiRNA- based promoter driven expression.
Major comments:
- Although the western blots in all the images do support the conclusions of each figure, the immunostaining is often missing and in the cases where it is presented, looks low resolution and not really discernable. In Figures 2, 3and 4, it will be important for the authors to show the immunostaining of claudin-1 and 3 in the SMG-C6 cell line , at probably different time points of treatment with high glucose, with corresponding stains for the appropriate controls like 0 hr and mannitol, and for claudins that did not change (4 or 7). This is especially significant on the transwells- they must be able to see the localization of claudins 1 and 3 and quantify them. Some images with the mouse SMG epithelial tissue and the Sp1 staining require more clarity.
- Fig 1. The db/m tight junctions look much more inwards from the apical than expected, perhaps the authors moved the labeling a bit further? what they point out as AJs might actually be gap junctions. Also, it is confusing that since barrier forming claudins 1 and 3 are actually enhanced which forms the basis of this study, the authors note that the tight junctions look discontinuous and obscure, rather it may possibly make the tight junctions more dense. If possible, TEM on the SMG-C6 cells might help elucidate the TJs more easily, which might overcome the complications of tissue preservation of mouse tissue that affects quality of EM.
- Immunohistochemistry and Fig 5 C , F the images and scale bars are pixelated and unclear and would benefit from being higher quality, co-localization indices must be measured.
- While the TER and 4KDa-Dextran flux appropriately reciprocate with each other, It is a bit challenging to understand the dramatic drop in paracellular permeability of 40 KDa Dextran. These are big molecules that are unable to pass through the tight junctions in general. A suggestion is to perhaps try the tracer on the apical side of the transwell and collect samples from the basal chamber? It is also possible that despite commercial dextrans often come in a range of molecular weights, and despite the average molecular weights written , the range of this mixture can vary.
Minor issues and comments:
2. It appears that the presence of glucose in the salivary epithelium poses a different phenotype than that of the intestinal epithelia, where glucose drives the absorption of both transcellular and paracellular sodium. This can be relavently discussed in the manuscript.
The authors should provide the source and catalog of all reagents used, some of which are missing, for eg the 4Kd and 40Kda dextrans.
This is also to note that the journal requires all the uncut, original blots to be presented untransformed or scaled as supplementary data.
Round 2
Reviewer 3 Report
Minor concern:
As with quantifications for all the western blots shown, it would be good to see the quantifications of the (representative) images shown in the supplementary data for the claudins 1, 3 and 4 immunofluorescence stains.
As the authors suggest, the closure of the cells completely to form tight junctions might be a reason for the drop in 40KDa dextran. It is suggested that to avoid inducing fluctuations, a longer equilibration time between changes in media composition might be better.
